# Research Advances on Molecular Mechanism of Salt Tolerance in *Suaeda*

**DOI:** 10.3390/biology11091273

**Published:** 2022-08-26

**Authors:** Wancong Yu, Wenwen Wu, Nan Zhang, Luping Wang, Yiheng Wang, Bo Wang, Qingkuo Lan, Yong Wang

**Affiliations:** 1Institute of Germplasm Resources and Biotechnology, Tianjin Academy of Agricultural Sciences, Tianjin 300384, China; 2Department of Agronomy, Tianjin Agricultural University, Tianjin 300392, China

**Keywords:** *Suaeda*, salt tolerance, ion and osmotic adjustment, antioxidant regulation, plant hormones secretion, photosynthetic system, omics mechanisms

## Abstract

**Simple Summary:**

This review comprehensively analyzes the molecular mechanism of *Suaeda* species under salt stress from aspects of physiology, biochemistry, transcriptomics, proteomics, and metabolomics, providing a theoretical basis for understanding the salt tolerance of *Suaeda*. The unique genetic and physiological characteristics of *Suaeda* support their high potential for utilization as promising biological resources to improve agriculture under saline conditions.

**Abstract:**

Plant growth and development are inevitably affected by various environmental factors. High salinity is the main factor leading to the reduction of cultivated land area, which seriously affects the growth and yield of plants. The genus *Suaeda* is a kind of euhalophyte herb, with seedlings that grow rapidly in moderately saline environments and can even survive in conditions of extreme salinity. Its fresh branches can be used as vegetables and the seed oil is rich in unsaturated fatty acids, which has important economic value and usually grows in a saline environment. This paper reviews the progress of research in recent years into the salt tolerance of several *Suaeda* species (for example, *S. salsa*, *S. japonica*, *S. glauca*, *S. corniculata*), focusing on ion regulation and compartmentation, osmotic regulation of organic solutes, antioxidant regulation, plant hormones, photosynthetic systems, and omics (transcriptomics, proteomics, and metabolomics). It helps us to understand the salt tolerance mechanism of the genus *Suaeda*, and provides a theoretical foundation for effectively improving crop resistance to salt stress environments.

## 1. Introduction

Soil salinization is the main limitation for agricultural economic development, and globally about 10% of soil is affected by high salinity [1]. Salt stress is an important type of abiotic stress, and can cause both osmotic and ionic toxicity in cells, seriously affecting the growth and yield of crops. However, as a halophyte, the genus *Suaeda* can survive and even grow healthily in high-salinity environments (salt concentrations of 200 mM or greater) [2,3,4]. Therefore, the investigation of the salt tolerance mechanism of the genus *Suaeda* will provide a molecular basis for better utilization of saline alkali land.

The genus *Suaeda*, an annual succulent herb of the Amaranthaceae/Chenopodiaceae [5], is a typical halophyte that usually grows in coastal, lakeside, desert, or swamp saline alkali environments [6]. Studies have shown that *Suaeda* seeds contain oil at approximately 20%, and are rich in unsaturated fatty acids, which have extremely high economic value and health benefits [7]. *Suaeda* species contain abundant protein, amino acids, minerals, and other essential micronutrients [8]. The tender seedlings are not only nutritious but also taste good [9]. In addition, the genus *Suaeda* has strong resistance to extreme environments such as cold, drought, and high salinity [10]. It can grow in desert alkaline soil and arid grassland, which means it has been considered a symbolic vegetation in maintaining the ecology of saline–alkali desert areas. As a natural polysalt plant, *Suaeda* absorbs soluble salt from saline soil to reduce the salt content of the soil [11]. Therefore, the genus *Suaeda* spp. is a priority plant for the reconstruction of saline and alkaline land.

In order to adapt to the stress of soil salinization and to reduce damage to their growth and reproduction, plants have evolved complex response mechanisms. These include, for instance, upregulating the genes and proteins that participate in salt tolerance, and promoting the production of phytohormones and metabolites that alleviate the toxic effects of salinity. In this review, we summarize the mechanisms of several *Suaeda* species response to salt stress. This review focuses on recent advances including ion regulation and compartmentation, osmotic adjustment of organic solutes, antioxidant regulation, plant hormones, changes in the pathway of photosynthetic system, transcription factors (transcriptomics), various stress-inducible proteins (proteomics), and the role of metabolites (metabolomics).

## 2. Ion Regulation and Compartmentation

Salt stress can disturb the ion balance in plants. Halophytes compartmentalize inorganic ions into the vacuolar cytoplasm mainly through transmembrane transport, thus increasing the osmotic pressure in the vacuolar, so that organelles can be protected from the toxic effects of ions, especially salt ions [12]. *Suaeda* species form an enhanced transmembrane ion gradient through tonoplast Na^+^/H^+^ antiporter (NHX), vacuolar membrane ATPase (V-H^+^-ATPase), vacuolar membrane proton pyrophosphatase (V-H^+^-PPase), K^+^ transporter, and chloride channels, maintaining the stability of Na^+^, K^+^, and Cl^−^ concentration, to protect *Suaeda* from salt ions and play an important role in the adapting to a high salt environment (Figure 1).

To relieve harmful accumulation of Na^+^, the Na^+^/H^+^ exchanger and V-H^+^-ATPase were upregulated in *Suaeda salsa* (L.) Pall. and *Suaeda maritima* (L.) Dumort., after NaCl treatment [13,14,15,16]. Similarly, the Na^+^ influx transporter of *Suaeda fruticosa* Forssk. [17], the plasma membrane H^+^-ATPase (PM-H^+^-ATPase) of *S. maritima* [15], and the V-H^+^-PPase in *S. salsa* leaves and roots [18] were significantly upregulated under saline conditions. Furthermore, the expression levels of V-H^+^-PPase genes *VP* in *S. salsa* and *S. corniculate* (C.A.Mey.) Bunge were upregulated in roots, stems, and leaves after salt stress induction. After transferring the *SsVP* gene into *Arabidopsis thaliana*, the activity of V-H^+^-PPase was significantly enhanced; overexpression of *ScVP* gene also increased the accumulation of Na^+^ in leaves and roots, lengthened the roots, and improved the salt tolerance of transgenic plants [19,20]. These results indicated that the enhanced activity of V-H^+^-PPase can translocate more Na^+^ to vacuoles, which plays an important role in protecting plants from salt ions and adapting to a high salt environment.

NHX1, high-affinity K^+^ Transporter (HKT), and Salt Overly Sensitive 1 (SOS1) are the major transporters involved in Na^+^ accumulation in plants [21]. Vacuolar NHX1 is a membrane protein that plays an important role in the exchange of Na^+^ for H^+^ across the vacuolar membrane, and the segregation of Na^+^ into vacuoles [22]. It was found that the tonoplast NHX1 was upregulated in *S. fruticosa*, *S. salsa*, *S. maritima*, and *S. corniculata* after salt stress [15,17,23,24]. After the *SsNHX1/SucNHX1* gene was transferred into rice, maize, and *Arabidopsis thaliana*, the salt tolerance of the transgenic plants was significantly improved [21,23,24,25]. Heterologous expression of the gene in poplars also enhanced the salt tolerance of transgenic poplar trees, which may result from the high expression of *SsNHX1/SucNHX1* gene, promoting Na^+^ accumulation in vacuoles, thus alleviating the effect of salt stress on cells [20,26]. These results suggested that *Suaeda* species might share similar mechanisms underlying ionic balance (Na^+^ and H^+^) in response to saline stress.

HKT1 and SOS1 are located on the cellular plasma membrane, and have opposite roles in controlling Na^+^ influx and efflux, respectively, across the plasma membranes of xylem parenchyma cells in the roots [21]. *SsHKT1* encodes an Na^+^-selective transporter that is preferentially expressed in the root xylem parenchyma and pericycle cells in *S. salsa* [21]. *SsSOS1* was highly expressed by high concentrations of NaCl (300 mM) in *S. salsa* roots [27]. *SsHKT1* coordinated with SsNHX1 and SsSOS1 to maintain Na^+^ accumulation under salt stress conditions by reducing Na^+^ retrieval from the xylem sap in *S. salsa* [21].

Although Na^+^ has been shown to suppress K^+^ influx in many plants, external Na^+^ treatment (25–400 mM) enhanced the growth of *Suaeda* species, including *Suaeda glauca* (Bunge) Bunge, *S. salsa*, *S. fruticosa*, and *S. maritima*; K^+^ concentrations in these plants were differential [28,29,30,31,32,33,34,35]. With increasing external NaCl concentrations, Na^+^ concentrations increased in the leaves of *S. fruticosa* [28] and *S. maritima* [32], K^+^ concentration decreased in *S. fruticos**a* leaves [28], and K^+^ content was relatively stable in *S. maritima* shoots [32,34,35]. *S. salsa* plant tissues can accumulate large amounts of Na^+^ and K^+^ under high salinity conditions [29,30,31]. For *S. glauca*, there was no competitive inhibition between Na^+^ and K^+^ absorptions [33]. Therefore, the role of Na^+^ and K^+^ in the *Suaeda* tissues may be to maintain water absorption by maintaining osmolality, which is essential for *Suaeda* plants’ survival in high-salinity conditions. Thus, the Na^+^ and K^+^ concentrations might be key requirements for growth of *Suaeda* species in highly saline soils.

Shao et al. [36] found that *SsHKT1;1*, a K^+^ transporter within *S. salsa*, was involved in salt tolerance by participating in cytosolic cation homeostasis, particularly mediating root K^+^ uptake and transport under salinity. HKT family and shaker AKT1-like channels in plants are considered the main channels that mediate K^+^ influx into root cells and correlate with salt tolerance [37,38,39]. The *SsAKT1* gene, encoding the inward rectifying K^+^ channel in *S. salsa*, significantly increased the transcript levels in roots with the increase of external Na^+^ concentration (25–250 mM) for 6 h [40]. Therefore, it may play an important role in salt tolerance of *S. salsa* by mediating both high- and low-affinity K^+^ uptake across different K^+^ concentration conditions and to the maintenance of K^+^ nutrition under salinity [40].

Although one HKT1-like transporter of *S. fruticosa* was found to be downregulated under salinity conditions, NHX, HKT, and PM-H^+^-ATPase showed no significantly different expression between salt-treated and control samples in *S. glauca* [41], suggesting that different *Suaeda* species may have different proteins regulating the ion balance. Therefore, *Suaeda* species may have certain similar and particular pathways that help them adapt to saline conditions.

In *Suaeda altissima* (L.) Pall., the expression of chloride channel (CLC) family genes *SaCLCc1*, *SaCLCd*, *SaCLCf* and *SaCLCg* in leaves increased with the increase of salt concentration, consistent with the accumulation of Cl^−^ in leaf cells. The results indicated that SaCLCc1, SaCLCd, SaCLCf and SaCLCg proteins may be involved in the separation of Cl^−^ in organelles, and may be involved in the mechanism of salt tolerance [42,43]. In addition, the complementation assay and bioinformatic analyses indicated that SaCLCc1 and SaCLCd proteins are Cl^−^/H^+^ antiporters, while the SaCLCf and SaCLCg proteins are likely Cl^−^ channels [42,43].

## 3. Osmotic Adjustment of Organic Solutes

In saline environments, halophytes can maintain intracellular osmotic balance by accumulating organic solutes such as sugars, alcohols, amino acids and their derivatives (proline, betaine, etc.), in addition to inorganic ions. These substances are soluble in water, are not toxicity, and do not intervene with cells’ biochemical reactions or various metabolic processes, even if accumulated at high concentrations. In this way, they not only protect the activity of enzymes in cells, but are also used as osmotic agents to maintain the osmotic balance of plant cells and improve the resistance of plants to salt stress [44].

Betaine is an important secondary metabolite that can be synthesized by cells for protection against osmotic stresses associated with high salinity [45]. In high-salt environments, plants store most of their NaCl in vacuoles through the Na^+^/H^+^ antiporter, and achieve equivalent osmotic potential by synthesizing compatible solutes such as glycine betaine in the cytoplasm. In the presence of NaCl, glycine betaine accumulated to maintain osmotic adjustment, playing an important role for *Suaeda* plants grown under high Na^+^ concentrations, inluding *S. salsa* [46], *S. maritima* [47], *S. fruticosa* [28], *Suaeda aralocaspica* (Bunge) Freitag & Schütze [48], *Suaeda eltonica* lljin [48], and *Suaeda heterophylla* (Kar. & Kir.) Bunge ex Boiss [48]. Thus, glycine betaine functions as an osmolyte to lower the plant’s water potential in order to protect membranes under high salinity conditions [48].

Under salinity conditions, proline also accumulated in *S. maritima* [47], *Suaeda physophora* Pall. [49], and *S. salsa* [50]. It was found that species with glycine betaine accumulators exhibited low proline content, and vice versa [51]. The enhancement of proline and glycine betaine may stimulate the expression of salt-tolerance proteins such as SKP1A in *S. maritima* [52]. In *S. salsa*, glycine betaine may play a more important role than proline in osmotic adjustment under high-salinity conditions [49]. Thus, as compatible solutes, betaine and proline appear to have different osmotic adjustment effects among *Suaeda* species.

Betaine is synthesized through converting choline into betaine aldehyde by choline monooxygenase (CMO), then catalyzing with betaine aldehyde dehydrogenase (BADH) [53]. Phosphoethanolamine methyltransferase gene *PEAMT* is related to betaine synthesis in *S. salsa* [54]. The *CMO* gene was upregulated in *S. salsa* [14,55,56], *Suaeda aegyptiaca* (Hasselq.) Zohary [57], and *S. maritima* [15] after salt stress, suggesting that *Suaeda* accumulates betaine to maintain osmotic balance. Under salt conditions, the *BADH* gene was induced to express in *S. salsa* [58,59], *S. maritima* [15], and *S. corniculate* [60] seedlings, which was involved in the biosynthesis of betaine to maintain the osmotic balance and enhance salt tolerance. Therefore, the *CMO, BADH*, and *PEAMT* genes involved in the synthesis of osmolytes are upregulated under salt stress. They may be positive regulators in response to NaCl [60], which can improve the tolerance of *Suaeda* cells to salt stress by maintaining the cells’ osmotic balance.

It has been reported that the expression levels of proline synthesis key enzyme gene *SsP5CS* (Δ1-dihydropyrrole-5 carboxylic acid synthase) and inositol synthesis key enzyme gene *SsINPS* are significantly increased in *S. salsa* in saline environments [61,62] Under high salinity (500 mM), *S. salsa* can accumulate organic acids, soluble sugars, lipid metabolites, and unsaturated fatty acids [63], as well as sucrose [56], helping the plant to deal with osmotic stress and increasing its nutritional value. However, in *S. corniculata*, the soluble sugars were downregulated after salt stress [64,65]. These results suggest that *Suaeda* plants enhance their resistance to osmotic stress by regulation of osmolytes under salinity conditions [56].

## 4. Antioxidant Capacity Regulation

Under salt-stress conditions, excessive reactive oxygen species (ROS) including hydrogen peroxide, hydroxyl radicals, and oxygen radicals are accumulated in plant cells, which can damage the cells’ macromolecules and membrane structures. Antioxidants such as superoxide dismutase (SOD), glutathione transferase (GST), ascorbic acid peroxidase (APX), glutathione reductase (GR), peroxidase reductase (PrxR), ascorbic acid glutathione (ASA)-glutathione (GSH) cycle enzyme, or catalase (CAT) can remove all kinds of free radicals and enhance the defense ability of cells against oxidative stress [66]. *Suaeda* was found to be able to maintain the balance between the formation and elimination of ROS by increasing the activity of antioxidant enzymes, for instance SOD [67,68], CAT [56,69], APX [69] and GPX [56].

With the increase of salt concentration, activities of Mn-SOD, Fe-SOD, and CuZn-SOD were detected in *S. salsa* leaves [67]. The activity of SOD in *S. salsa* and *S. maritima* increased significantly under saline conditions [67,68]. In medium containing 400 mM NaCl, the SOD activity of *Suaeda japonica* Makino leaves increased, resulting from the concentration of the substrate superoxide anion and the production of O_2_^•−^ under salt stress [70,71].

*GST* gene expression levels were greatly increased in roots of *S. maritima* upon salt treatment [72]. However, in *S. fruticosa*, the level of glutathione increased with high salt (900 mM) treatment, and also with 0 mM NaCl treatment, while it decreased with 300 mM NaCl [73]. After transforming *Arabidopsis* with *S. salsa*’s *GST* gene, the salt tolerance of transgenic *Arabidopsis* was improved, possibly due to the overexpression of the *S. salsa GST* gene in *Arabidopsis* plants alleviating the effect of reactive oxygen free radicals and enhancing the tolerance of cells to salt stress [74].

Dehydrins (DHN), known to be chaperones, could bind to the hydrophilic sites of proteins to scavenge oxygen free radicals, thereby reducing the peroxidation damage caused by stress conditions and enhancing the resistance of plants [75,76]. The expression of the dehydrin gene *SsDHN* in *S. salsa* is induced by salt stress [77]. Overexpression of the *S. glauca DHN* gene in yeast could enhance tolerance to salt stress [78]. These results indicated that *S. salsa* can scavenge various reactive oxygen free radicals through antioxidant substances, and enhance the tolerance of cells to salt stress, enabling adaptation to the saline–alkali environment.

After the application of NaCl, the activity of CAT increased significantly in *S. salsa* and *S. maritima* [68,69]. The expression levels of the *Sscat1* and *APX* genes in *S. salsa* increased significantly with salt stress [79,80]. Overexpression of the *APX* gene in *Arabidopsis* can increase *APX* activity, lower the H_2_O_2_ content, and reduce cell membrane damage caused by salt stress [3,81].

It has also been reported that in *S. salsa* after treatment with 200 mmol/L NaCl for seven days, the activities of GR in the chloroplast matrix and thylakoid were increased, along with ASA and GSH content, while H_2_O_2_ content and membrane lipid peroxidation decreased [69]. These results indicated that the increase of GR activity promoted the production of GSH, enabling the scavenging of reactive oxygen species. Therefore, increased GR content may be an important reason for the decrease of H_2_O_2_ content in *S. salsa* leaves [67].

In conclusion, by increasing the activity of its antioxidant enzymes, *Suaeda* can maintain the balance between the formation and elimination of ROS. Plant antioxidant systems are generally classified into enzymatic and non-enzymatic systems. The enzymatic defense systems in *Suaeda* include SOD [67,68], CAT [56,69], APX [69], and GPX [56]. Non-enzymatic defense systems in *Suaeda* are AsA and GSH [71], etc. Under salinity conditions, the increase of ROS results in the relatively inadequate antioxidant scavenging capacity of antioxidant enzymes, leading to oxidative stress. *Suaeda* scavenges ROS produced by salt stress mainly through the synergy of SOD activation, the CAT, GPX, and PrxR pathways, and the ASA-GSH cycle enzyme [71]. Therefore, the activities of various enzymes and the content of non-enzyme substances involved in the process of scavenging H_2_O_2_ can reflect the salt resistance of *Suaeda* under salt stress.

## 5. Secretion of Plant Hormones

Phytohormones integrate various signals in maintaining responses to salt stress and other stresses [82]. Salt stress affects the reproductive growth and yield of plants by regulating the secretion of plant hormones including indole acetic acid (IAA), gibberellin (GA), cytokinins (CTK), ethylene (ETH), abscisic acid (ABA), etc., but promotes the reproductive growth of euhalophytes. CTK and IAA can increase the salt tolerance of seeds in *Arabidopsis* [62]. Under salt stress, the inhibition of hormone synthase activity led to reduction or cessation of IAA and CTK synthesis, which delayed plant growth, but increased the content of ABA and ETH in rapeseed [83].

Exogenous ABA pretreatment can increase chlorophyll pigment content and accumulation of inorganic osmolytes, thus reducing the damage of salt stress and increasing the general growth rate of *S. maritima* [52]. Salt stress can also induce the accumulation of ABA in *S. salsa* seeds. Under salt stress, 1-aminocyclopropane-1-carboxylate (ACC, the direct precursor of ethylene), GA_4_, and 6-benzyladenine (BA) can promote the germination of seeds, which indicates that these hormones can reduce the impact of salt stress on seeds and improve the ecological adaptability of *S. salsa*, *S. maritima*, and *Suaeda prostrata* Pall. seeds to a saline alkali environment [84,85,86]. High salinity inhibited seed germination by decreasing the levels of GA4 in *S. salsa* [85]. During the later stages of plant vegetative growth, NaCl treatment can significantly increase the content of endogenous GA3, GA4, ABA, and brassinolide (BR) in the stems of *S. salsa* plants [87]. In the flowering stage, NaCl treatment significantly increased the content of GA_3_, GA_4_, IAA, and zeatin (ZR) in floral organs compared with the control [87]. In response to salt stress, ETH-related pathways are upregulated in *S. glauca* and *S. maritima* [15,41]. The auxin, ETH, and jasmonic acid (JA) signaling transduction pathways were all upregulated in *S. salsa* after saline treatment, and are important to gene regulations of ion transport and antioxidation [87]. In addition, genes related to the biosynthesis of ZR, IAA, GA, BR, and ABA, and to plant hormone signal transduction, including genes encoding CYP735A, CYP85A, GID1, NCED, PIF4, AHP, TCH4, SnRK2, and ABF, were upregulated in *S. salsa* treated with NaCl. Downregulation of gibberellin 2-oxidase 2 was observed in *S. fruticose* after 300 mM salt treatment [17].

Some plant hormones positively regulate salt tolerance, while others play a negative role. GA3 was found to stimulate growth at all salinities for *S. maritima* and *S. rostrata*, while kinetin (KT) proved to be inhibitory to plant growth at higher salinities [86]. These results indicate that the synergistic upregulation of genes involved in plant hormone synthesis and signal transduction contributes to the reproductive growth of *S. salsa* under salt stress [87]. Therefore, in response to salt stress, plants build a defense system by orchestrating the synthesis and signaling pathways of various hormones via multiple crosstalks.

## 6. Changes in the Pathway of Photosynthetic System

Halophytes can fit into or resist the influence of saline environments by regulating photosynthesis and metabolism. *Suaeda* is a kind of halophyte that grows in high salt environments and extreme high-tide zones. Photosynthesis plays an important role in the accumulation of protein biomass within halophytes in saline soil. The halophytes *Suaeda* (Chenopodiaceae) include species with both C3 and C4 photosynthetic pathways [88].

High salt stress (200–500 mM NaCl) prompted decent protection of the light response system in *S. salsa*, maintaining the structure of the light system, promoting light-energy transmission, and improving the activities of related enzymes [63,89]. After 200 mM NaCl treatment, the expression of carbon-assimilation-related enzyme genes *SsFNR*, *SsRbcl*, *SsRbcs*, *SsRCA*, *SsPGK*, and *SsGAPDH* increased significantly, indicating that carbon-assimilation-related enzymes may play an important role in promoting the photosynthesis of *S. salsa* [63].

Under salinity conditions, the concentration of photosynthetic pigments chlorophyll a, chlorophyll b, and total chlorophyll in the leaves of *Suaeda schimperi* Moq., *Suaeda vermiculata* Forssk. Ex J.F.Gmel., *Suaeda monoica* Forssk. Ex J.F.Gmel. were different [90]. The chlorophyll content in *S. salsa* was improved after NaCl stress [90]. When cultured with 200 mM NaCl for 14 days, the photosynthetic capacity including net photosynthetic rate (Pn), electron transfer rate, NADPH level, activities of ferredoxin-NADP oxidoreductase, ribulose-1, 5-bisphosphate carboxylase (Rubisco), and Rubisco activase were improved significantly in *S. salsa*, and 500 mM NaCl had no adverse effect on those parameters [31,90]. Low salt stress had little effect on the photosynthesis of *S. corniculata* seedlings, while high salinity inhibited their photosynthesis [91]. The maximum photochemical quantum yield Fv/Fm of photosystem II (PSII), the photochemical quenching coefficient qP, and the non-cyclic photosynthetic electron transfer rate RE, T of PSII were not affected by low salt stress, but showed a downward trend in *S. corniculata*, *S. salsa*, and *S. aegyptiaca* [58,91,92]. All these trends indicate that the seedlings of *Suaeda* can adapt to a high salt environment by changing their rates of photosynthesis.

*SsPsaH* is a member of the H subunit of the PSI reaction center in *S. salsa*, and its expression level increases with salt stress [93]. Overexpression of *SsPsaH* in recombinant yeast can enhance the tolerance of transformants to salt stress [93]. The expression of *S. salsa* glycerol3-phosphate acyltransferase (*SsGPAT*) was also increased by salt stress; compared with the wild type, high salinity prompted a smaller reduction in chlorophyll content, PSII photochemical efficiency, photosystem I (PSI) redox activity (δI/Io), and unsaturated fatty acid content of phosphatidylglycerol (PG) in *Arabidopsis* thaliana seedlings. This indicates that the overexpression of the *SsGPAT* gene in *Arabidopsis* can enhance the salt tolerance of PSII and PSI under salt stress by upregulation of the unsaturated fatty acid content of PG, thereby alleviating the photoinhibition of PSII and PSI [94]. PS II in *S*. *salsa* shows a high resistance to low salinity [92], making plant growth more adaptable under low salinity conditions (8 ppt). Therefore, increased photosynthetic activity may play a critical role in the biomass enhancement of *Suaeda* under saline conditions [90].

## 7. Omics Approaches

Omics approaches, i.e., transcriptomics, proteomics, and metabolomics, play an important role in the study of plant salt tolerance [95]. Omics techniques, including characterization of transcription factors, proteins and metabolites involved in salt tolerance, have been applied to understand response mechanisms in plants and utilized for generating salt-tolerant crops.

### 7.1. Transcriptomics

The responses of plants to salt stress trigger multi-factor synergistic effects. The related transcription factor can regulate the expression of a series of genes, thereby enhancing the resistance of plants to the saline environment. The regulation of transcription factors in genes can be realized through specific binding with cis elements to initiate the transcription expression of the gene [63,96]. According to the different DNA binding domains, plant transcription factors can be divided into different families. Among these, the transcription factor families related to salt-stress response mainly include NAC, AP2/EREBP, HB, MYB, BZIP/HD-Zip, and WRKY. Under saline conditions, transcription factors can be regarded as the virtual switches that directly upregulate or downregulate the expression of salt-stress-related genes.

Transcriptome analysis of *S. salsa* showed that HB, MYB, and bZIP transcription factors were regulated by salt stress, and the transcriptional regulation of HB-7 and MYB78 was found to alleviate the damage of salt stress in *S. salsa* [97]. In *S. salsa* leaves, MYC2 was significantly upregulated after saline treatment compared with the control [87]. RNA sequencing analysis revealed that WRKY and bHLH transcription factors involved in salt tolerance were upregulated in *S. glauca* and *S. rigida* [41,98,99]. *MYB* genes were also elevated in *S. maritima* and *S. glauca* in response to salt stress [15,100]. Study of the transcription factors showed that MYB07, MYB37, and BZIP59 played important roles for regulation of salt tolerance in *S. fruticosa* [101].

Salt stress can induce the expression of transcription factor genes *SlNAC1*, *SlNAC2*, *SlNAC7*, and *SlNAC8* in *S. salsa*. Compared with wild-type *Arabidopsis*, salt stress can promote the germination and survival rates of *SlNAC1* and *SlNAC8* transgenic *Arabidopsis*, respectively, but was found to inhibit root growth in the transformants [102,103,104]. Overexpression of *SlNAC2* and *SlNAC7* in *Arabidopsis* can enhance tolerance to salt stress [103,105]. Moreover, the overexpression of *SlNAC8* in transgenic plants also enhanced the expression of stress response genes *RD20*, *GSTF6*, *COR47*, *RD29A*, *RD29B*, and *NYC1* [104]. In brief, *SlNAC1*, *SlNAC2, SlNAC7*, and *SlNAC8* transcription factors may make contributions to changes in the physiological and biochemical characteristics of plants by regulating the expression of stress-responsive genes, thus enhancing the resistance of plants to salt stress.

The expression levels of *AP2* were upregulated in germinating seeds of *S. glauca* when exposed to different NaCl concentrations [100]. In *S. salsa* leaves, *ERF1/2* were significantly upregulated by saline treatment compared with the control [87]. In *S. salsa*, DREB protein belongs to the CBF/DREB transcription factor, and salt stress can significantly promote *SsDREB* gene expression. Overexpression of the *SsDREB* gene in tobacco can enhance the salt tolerance of transgenic plants [106]. Two *DREB* genes of *S. salsa* can respond to high salt stress through independent ABA pathways, while *SsCBF1* may be involved in the regulation of high salt stress through ABA signaling [107,108].

In short, transcription factors play a key role in the responses of plants to salt stress. By overexpressing transcription factor genes in transgenic plants, the expression of transcription-factor-specific binding genes can be regulated, in order to obtain stronger salt tolerance. Therefore, salt-stress-responsive transcription factors can be used as an important tool for the genetic engineering of plants’ salt tolerance.

### 7.2. Proteomics

Proteomics has become a very important technique in the post-genomic era [109], serving as a powerful tool for describing complete protein changes at organ, tissue, cell, and organelle levels under various stress conditions [110]. Therefore, proteomic investigation can reveal the potential associations between protein expression and plant stress acclimation. Proteomic methods have been widely used for investigating specific genes and proteins contributing to salt tolerance and survival in saline conditions [57,67].

The salinity-responsive proteins belong to various functions including ROS scavenging, ionic and osmotic regulation, signal transduction, and photosynthesis [111]. In *S. maritima*, photosynthesis, heat shock proteins, peroxidase, expansins, signaling processes, and modulation of transcription/translation were modulated by salinity [112]. In *S. salsa*, three upregulated and six downregulated proteins were identified, involved in photosynthesis, energy metabolism, stress, and defense [109]. The metabolism proteins of *S. salsa* were involved in the pentose phosphate pathway, polyamine biosynthesis, amino acid biosynthesis, and isoprenoid biosynthetic processes [109]. In *S. corniculate*, 10 proteins were observed as being differentially expressed under NaCl treatment [64]. According to KEGG pathway analysis, these proteins were involved in carbohydrate metabolism, energy metabolism, photosynthesis, nucleotide metabolism, protein synthesis, stress, and defense, or were unknown [64].

Comparative proteomic analysis was used to determine the proteomic profiles of *S. salsa* exposed to salinity [109]. The ATP synthase CF1 alpha subunit and ATPase subunit were downregulated by salinity exposure, implying disturbance in energy metabolism [109]. Salt treatment upregulated plasma membrane aquaporins [113], suggesting that *S. salsa* maintains homeostasis and ion distribution by increasing leaf succulence and compartmenting the ions. Mitogen-activated protein kinase (MPK6), ethylene-insensitive protein 2 (EIN2), and ethylene-insensitive protein 3 (EIN3) were significantly upregulated in *S. salsa* leaves after saline treatment compared with the control [87]; the result was consistent with the observation that ethylene signaling is indispensable for tolerance to saline stress in plants [114].

Protein pattern analysis revealed that 22 kDa and 55 kDa proteins occurred in salt-treated *S. maritima* leaves [52]. The enhancement of nontoxic metabolites may stimulate the expression of salt-tolerance proteins in *S. maritima* [52]. Acetolactate synthase 1 and histone H4 were up- and down-accumulated in *S. maritima* at the lower (200 mM) and higher (500 mM) NaCl dosages, respectively [112]. In *S. maritima*, cytochrome b6f complex, cytosolic, expansin-B1, chloroplastic GAPDH, and the chloroplastic ATP synthase subunit α were downregulated with salt treatment, resulting in a reduction of photosynthetic activity [112]. Thus, proteomics is helpful to understand the characterization of interactions and the response to salt stress in *Suaeda* plants.

### 7.3. Metabolomics

Metabolites, compatible solutes, and bioactive compounds are biomolecules produced by plants under natural or stressed conditions. Under stress conditions, plant systems need to regulate metabolite levels to maintain basal metabolism and achieve homeostasis [115]. Metabolomics is a potent approach to the identification and quantification of all low-molecular weight metabolites required by plants in response to abiotic stress. It can be used to study metabolic pathways or metabolites associated with salt-stress tolerance [116]. In halophytes, metabolites involved in salt tolerance include glycine betaine, proline, pinitol, mannitol, sorbitol, O-methyl muco-inositol, inositol, and polyamines [116].

In plant metabolomics, metabolites are divided into primary and secondary categories. During plant stress, primary metabolites directly contribute to the accumulation of compatible solutes, such as amino acids, sugars, and sugar alcohols, to cope with osmotic stress [117]. Under saline conditions, 61 primary metabolites were detected in the leaves of *S. monoica* and *S. fruticosa* species [117]. These metabolites included amino acids, sugars, sugar acids, fatty acids, different compounds, and flavonoid (kaempferol) groups [117]. Serine concentration was higher in *S. monoica* compared with *S. fruticosa* under saline conditions [117]. In *S. corniculate*, 21 metabolites were identified under conditions of salinity, including amino acids, carbohydrates, organic osmolyte, and intermediates in the tricarboxylic acid (TCA) cycle, among others (e.g., ethanol, dimethylamine O-phosphocholine, and choline) [65].

Under saline conditions, amino acids including valine, alanine, glutamate, tyrosine, leucine, isoleucine, and phenylalanine were decreased in the aboveground parts of *S. salsa* and *S. corniculata* seedlings [58,64]. Furthermore, the total protein content in the aboveground parts of *S. salsa* seedlings decreased with increasing concentrations of salinity (0 mM–170 mM–500 mM) [58]. However, in the *S. salsa* root tissues, the metabolic responses were different from the aboveground parts of the seedlings. The proline and citrate in the root tissues were uniquely increased, and the branched-chain amino acids, lactate, choline, phosphocholine, glutamine, and fructose were uniquely decreased. The differences of metabolic responses between the roots and aboveground parts of *S. salsa* seedlings suggest different regulating mechanisms in various tissues under salt treatment [58]. Moreover, studies have shown that high salinity can lead to accumulation of more amino acids (500 mM > 200 mM > 0 mM) in *S. salsa* [63], and increases in leucine, isoleucine, valine, and glutamine in *S. corniculata* seedlings [65].

As osmolytes, betaine and proline can be synthesized for protection against salinity stress [45]. In *S. salsa* [58] and *S. corniculata* [64], betaine was significantly elevated in the aboveground parts of seedlings under salt stress, and *CMO* was induced in *S. salsa* [55] and *S. aegyptiaca* [57] after salt stress, suggesting that *Suaeda* accumulates betaine to maintain osmotic balance. The *Suaeda* species exhibited significant variation in amino acid biosynthesis under similar salinity conditions. For example, *S. monoica* and *S. schimperi* accumulated significantly higher foliar proline than *S. vermiculata*, suggesting that proline is an important compatible osmolyte in *S. monoica* and *S. schimperi* species [90].

The metabolic processes of *Suaeda* include primary and secondary metabolism, and the accumulation of metabolites is related to the osmotic tolerance, energy supply, and nutritional value of *Suaeda* species. Changes in metabolite content may play an important role in maintaining cell osmotic potential, protecting cell membrane structure, and promoting resistance by destruction of ROS. The related studies showed that there were differences in the accumulation of metabolites among *Suaeda* species under different salt-stress conditions. Therefore, the metabolic mechanisms of compounds in *Suaeda* species are complex and deserve to be further investigated.

## 8. Conclusions and Perspectives

Salinization is an important issue for global agricultural productivity and food security, which seriously affects crop growth and yield. Improving crops’ salt tolerance is the most direct and effective way to solve this problem. As halophytes, the genus *Suaeda* can grow healthily in high salt environments. Over twenty species of *Suaeda* have been described for their ability to survive under high salinity environments [64]. For example, *S. salsa* grew equally well with 400 mM NaCl as with 10 mM NaCl, and its growth optimal concentration was found to be 200 mM NaCl [31]. Furthermore, *Suaeda* seedlings have important economic value, the fresh branches can be used as vegetables, and its seed oil is edible and rich in unsaturated fatty acids [3]. Therefore, *Suaeda* is a prime salt-tolerant model plant with great economic value. Elucidating the salt-tolerance mechanism of *Suaeda* is helpful for developing salt-tolerant plant varieties and making effective use of saline land resources.

The salt-tolerance mechanism of the genus *Suaeda* is very complicated, involving cells, tissues, and organs, and the whole plant, being the result of the synergy of plant physiology, biochemistry, molecular, transcript, protein, and metabolic level (Figure 2). This review has analyzed the genes (Table 1), enzymes, proteins, and metabolites that participate in salinity adaption, and investigated the roles of these factors in *Suaeda* ion transport (Figure 1), osmotic regulation, free radical scavenging, hormone regulation, and photosystem regulation, thereby summarizing the molecular mechanisms of salt tolerance in *Suaeda* (Figure 2). At present, a series of genes related to salt tolerance have been cloned and preliminary functional verification has been carried out. Further research should be undertaken to consider methods of fully improving adaptable properties in terms of salt-stress response, and coordinatively regulating the multiple salt-tolerance genes in the genus *Suaeda*. The systematic combination of various “omics”, including genomics, transcriptomics, proteomics, and metabolomics, is necessary to understand the molecular networks underlying *Suaeda* species’ responses to salt stress. The integrated application of multiple “omics” technologies and precise genome editing by the CRISPR/Cas9 system can be further conducted in future studies, which will lay a foundation for elucidating the molecular regulation mechanism of the genus *Suaeda*, to enable further salt-tolerant plant breeding. Overall, this review might enhance an integrated comprehensive understanding of salt tolerance. These results may provide elite genetic resources for the modification of salinity-resistant crop species, and improve the efficiency of saline–alkali land utilization.

## Figures and Tables

**Figure 1 biology-11-01273-f001:**
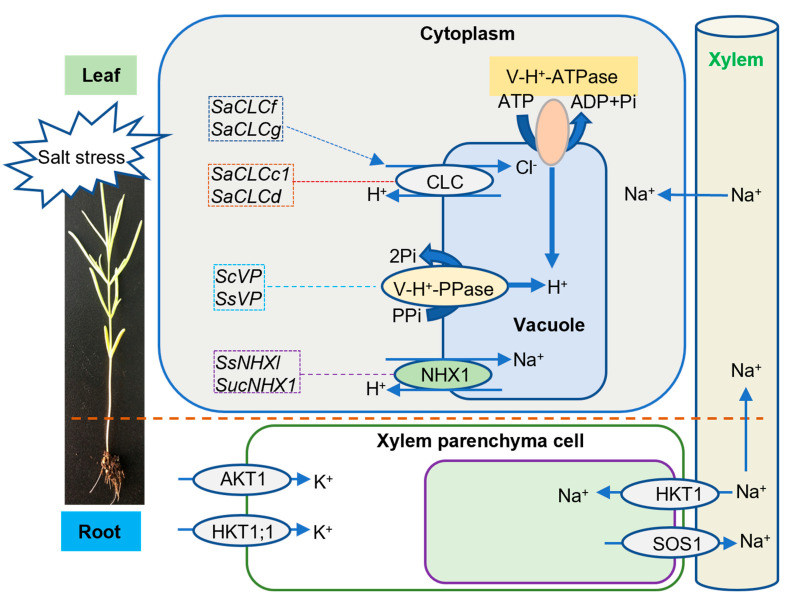
Schematic diagram of transmembrane transporters transporting Na^+^, H^+^, K^+^, and Cl^−^ in *Suaeda*.

**Figure 2 biology-11-01273-f002:**
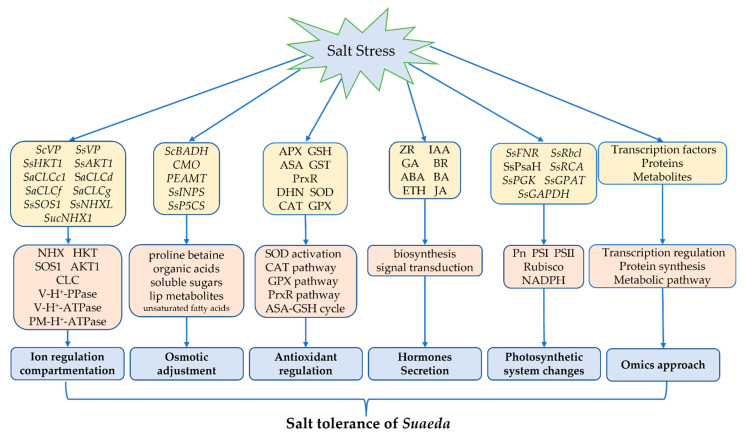
Researched advances of molecular mechanism in the genus *Suaeda*.

**Table 1 biology-11-01273-t001:** The genes in *Suaeda* species and their probable functions.

*Suaeda* Species	Gene	Function	Reference
*Suaeda salsa*	*SsVP*	Encoding V-H^+^-ATPase	[20]
	*SsNHX1*	Na^+^/H^+^ antiporter	[23]
	*SsSOS1*	Na^+^/H^+^ antiporter	[27]
	*SsHKT1;1*	K^+^ uptake under salt stress	[21]
	*SsHKT1*	K^+^ uptake under salt stress	[36]
	*SsAKT1*	K^+^ uptake under salt stress	[40]
	*SsCMO*	Encoding choline monooxygenase	[56]
	*SsBADH*	Encoding betaine aldehyde dehydrogenase	[56]
	*SsP5CS*	Encoding D1-pyrroline-5-carboxylate synthase	[61]
	*SsINPS*	Encoding myo-inositol-1-phosphate (I-1-P) synthase	[62]
	*Sscat1*	Encoding catalase	[80]
	*SsAPX*	Encoding ascorbate peroxidase	[79]
	*SsGST*	Encoding glutathione S-transferase	[74]
	*SsDHN*	Encoding betaine aldehyde dehydrogenase	[77]
	*SsFNR*	Encoding carbon-assimilation-related enzymes	[63]
	*SsRbcl*	Encoding carbon-assimilation-related enzymes	[63]
	*SsRbcs*	Encoding carbon-assimilation-related enzymes	[63]
	*SsRCA*	Encoding carbon-assimilation-related enzymes	[63]
	*SsPGK*	Encoding carbon-assimilation-related enzymes	[63]
	*SsGAPDH*	Encoding carbon-assimilation-related enzymes	[63]
	*SsGPAT*	Encoding glycerol3-phosphate acyltransferase	[95]
	*SsCBF1*	CBF/DREB transcription factor	[109]
	*ERF1/2*	ERF transcription factor	[88]
*S. corniculate*	*ScVP*	Encoding V-H^+^-ATPase	[19]
	*SucNHX1*	Na^+^/H^+^ antiporter	[24]
	*ScBADH*	Encoding betaine aldehyde dehydrogenase	[60]
*S. liaotungensis*	*SlCMO*	Encoding choline monooxygenase	[59]
	*SlBADH*	Encoding betaine aldehyde dehydrogenase	[57]
	*SlPEAMT*	Encoding phosphoethanolamine methyltransferase	[54]
	*SlNAC1*	NAC transcription factor	[102]
	*SlNAC2*	NAC transcription factor	[102]
	*SlNAC7*	NAC transcription factor	[105]
	*SlNAC8*	NAC transcription factor	[102]
*S. maritima*	*SmGST*	Encoding glutathione S-transferase	[72]
	*CMO*	Encoding choline monooxygenase	[15]
	*BADH*	Encoding betaine aldehyde dehydrogenase	[15]
*S. glauca*	*SgDHN*	Encoding betaine aldehyde dehydrogenase	[78]
	*AP2*	AP2 transcription factor	[100]
*S. altissima*	*SaCLCc1*	Encoding chloride channel protein	[43]
	*SaCLCd*	Encoding chloride channel protein	[42]
	*SaCLCf*	Encoding chloride channel protein	[42]
	*SaCLCg*	Encoding chloride channel protein	[42]
*S. aegyptiaca*	*CMO*	Encoding choline monooxygenase	[58]

## Data Availability

Not applicable.

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
