# Peer review of "Research Advances on Molecular Mechanism of Salt Tolerance in Suaeda"

_biology, 2022, doi:10.3390/biology11091273_

Round 1

Reviewer 1 Report

This review is devoted to advances in investigations of molecular mechanisms, biochemistry, and physiology of halophytes related to Suaeda genus. In two last decades, the representatives of this genus widely used as modelling plants for study processes underlaying salinity tolerance. The appearance of such a review is certainly useful and timely. The authors were able to review the available works covering the main aspects of molecular biology, biochemistry, and physiology of the genus Suaeda and to consider the current approaches to the study of these plants. Unfortunately, the manuscript has many disadvantages, which, however, can be eliminated with a thorough revision.

The manuscript is generally superficial, lacking depth of insight, bad literature style and bad English

Below, the main and minor comments are given in one list along text.

1)      Lines 65-67. Here confusion in the mechanisms of protection against the toxic and osmotic effects of salt. To avoid toxic effects cells transport ions from cytoplasm into vacuoles and by this increase osmotic pressure in vacuoles (decrease osmotic potential) what contributes to avoiding osmotic effect.

2)      Correctly: Suaeda fruticosa, «a» must be at the end of the word

3)      Lines 101-118. Legend to Figure 1. Very wordy and not very clear. In relation of ion transport, it is hardly understandable, where the authors mean the level of cell and where of whole plant. The same, about transport across plasma membrane and vacuole. In what part of the plant the content of ions rises and in what part decreases? Speaking of ion transport, it is necessary to clearly distinguish between the cellular level and the level of the whole plant.

4)      Figure 1. Unclear, why are CLC, VP and NHX1 related to nucleus? SOS1 and HKT1 are PM proteins, in the figure in low right part, these proteins located rather in tonoplast. The Figure and the legend to it need in clarifying and more accuracy. Speaking about Cl-/H+ antiporters of CLC family in S. altissima, it is not correctly to refer to SaCLCg and SaCLCf, because these prateins contain only one conserved glutamate. SaCLCc1 and SaCLCd are most probably Cl-/H+ antiporters.

5)      Line 138. Betaines are not alkaloids.

6)      Lines 139-147. Betaines are not only osmoregulators, but they have many other functions. In paticular, it should be noted their protective function. They perform the function of osmoregulation in the cytoplasm, in vacuoles of halophytes, ions mainly increase the osmotic pressure.

7)      Lines 186-187. Unclear. In response to what did activity increase in control?

8)      Lines 189 -191. Not exact wording. Due to overexpression of the A. thaliana gene or of expression of the gene from S. altissima?

9)      Line 201. This paragraph is about enzymes scavenging ROSs, but not about MDA.

10)   Lines 214-215. Why inactivation? This contradicts the above sentence. Indeed, SOD activity increases in response to salt treatment and triggers cascade of the reactions leading to ROS scavenging. This paragraph needs a scheme outlining the system of ROS scavenging enzymes and nonenzymatic antioxidant substances, as well as about effects of NaCl. 

11)   Lines 224-225. It is necessary to clearly distinguish between what is known about hormones in Arabidopsis and other glycophytes under salinity and available data from halophytes.

12)    Not all abbreviations are deciphered, for example, for Ba, CTK, Z, they are absent.

13)   Lines 219-242. In general, the Paragraph «Secretion of Plant Hormones» is written very superficially and sparingly.

14)   Lines 246-247. Not only through photosynthesis.

15)   Lines: 248-251; 255-257; 262-264; 270-272: 307-310; 319-322; 326-328. Bad English and style.

16)   Lines 273-276. Not clear what was compared to what. In general. bad sentence.

17)   Lines 350-354. Logic seems to be absent in this sentence. It is necessary to make it clearer.

18)   Lines 375-377. It is not necessary that decrease in content of an intermediate in the TCA cycle indicates an inhibition of the metabolic pathway as a whole because the metabolite content depends on both its input and its output rates.

19)   Figure 2 needs improvement. The second row, marked by pink (processes) needs to make in more details. Besides, the right column (Transcriptomics...) is not a process, but a research approach, so it does not fit well to the other columns outlining processes.

20)   Lines 400-407. Bad English. Besides, what means Regionalization? I think the authors mean Compartmentation.

21)   There is a mistake at the bottom of Table 2, a shift of column Function in relation to column Gene.

The manuscript needs in major revision to be published.

Author Response

1) Lines 65-67. Here confusion in the mechanisms of protection against the toxic and osmotic effects of salt. To avoid toxic effects cells transport ions from cytoplasm into vacuoles and by this increase osmotic pressure in vacuoles (decrease osmotic potential) what contributes to avoiding osmotic effect.

Response: Thank you for your comment. We have modified the sentence according to your suggestions. The revised sentence can be found in Line 65-68.

2) Correctly: Suaeda fruticosa, «a» must be at the end of the word

Response: Thank you for your suggestion. We have now corrected the word “Suaeda fruticose” to “Suaeda fruticosa”.

3) Lines 101-118. Legend to Figure 1. Very wordy and not very clear. In relation of ion transport, it is hardly understandable, where the authors mean the level of cell and where of whole plant. The same, about transport across plasma membrane and vacuole. In what part of the plant the content of ions rises and in what part decreases? Speaking of ion transport, it is necessary to clearly distinguish between the cellular level and the level of the whole plant.

Response: Thank you for your comments. We have revised the ions content in plant different part according to your suggestions. The revised sentence can be found in Line 113-122.

Figure legend 1 has revised as Line 101-102.

4) Figure 1. Unclear, why are CLC, VP and NHX1 related to nucleus? SOS1 and HKT1 are PM proteins, in the figure in low right part, these proteins located rather in tonoplast. The Figure and the legend to it need in clarifying and more accuracy. Speaking about Cl-/H+ antiporters of CLC family in S. altissima, it is not correctly to refer to SaCLCg and SaCLCf, because these prateins contain only one conserved glutamate. SaCLCc1 and SaCLCd are most probably Cl-/H+ antiporters.

Response: Thank you for pointing out the mistake of CLC, VP, NHX1 related to nucleus. We have revised it in Figure 1.

According to the reference 21 and 27, SsHKT1;1 and SsSOS1 are preferentially expressed in roots of S. salsa, so, they were figured in root part.

  1. Wang, W.; Liu, Y.; Duan, H. Yin.; X. Cui, Y.; Chai, W.; Song, X.; Flowers, T.; Wang, S. SsHKT1;1 is coordinated with SsSOS1 and SsNHX1 to regulate Na+homeostasis in Suaeda salsa under saline conditions. Plant Soil 2020, 449, 117–131.
  2. Liu, Q.; Liu, R.; Ma, Y.; Song, J. Physiological and molecular evidence for Na+ and cl exclusion in the roots of two Suaeda salsa Aquat. Bot. 2018, 146, 1–7.

We have revised the mistake of SaCLCg and SaCLCf in Figure 1, and SaCLCc1 [43] was added as a Cl-/H+ antiporters.

  1. Nedelyaeva, O.I.; Shuvalov, A.V.; Mayorova, O.V.; Yurchenko, A.A.; Popova, L.G.; Balnokin, Y.V.; Karpichev, I.V. Cloning and functional analysis of SaCLCc1, a gene belonging to the chloride channel family (CLC), from the halophyte Suaeda altissima (L.) Pall. Dokl. Biochem. Biophys. 2018, 481, 186–189.

5) Line 138. Betaines are not alkaloids.

Response: Thank you for pointing out this mistake. We have revised the sentence.

6) Lines 139-147. Betaines are not only osmoregulators, but they have many other functions. In paticular, it should be noted their protective function. They perform the function of osmoregulation in the cytoplasm, in vacuoles of halophytes, ions mainly increase the osmotic pressure.

Response: Thank you for your comment. We have revised the sentence according to your suggestions and deleted the section of betaines function just as Line 157-166.

7) Lines 186-187. Unclear. In response to what did activity increase in control?

Response: Thank you for your comment. We have revised the control to 0 mM NaCl treatment.

8) Lines 189 -191. Not exact wording. Due to overexpression of the A. thaliana gene or of expression of the gene from S. altissima?

Response: Thank you for your comment. We have revised the sentence in Line 218.

9) Line 201. This paragraph is about enzymes scavenging ROSs, but not about MDA.

Response: Thank you for pointing out this mistake. We have deleted the part about MDA.

10) Lines 214-215. Why inactivation? This contradicts the above sentence. Indeed, SOD activity increases in response to salt treatment and triggers cascade of the reactions leading to ROS scavenging. This paragraph needs a scheme outlining the system of ROS scavenging enzymes and nonenzymatic antioxidant substances, as well as about effects of NaCl.

Response: Thank you for your suggestion and pointing out the mistake. We have revised it and modified the sentence as your suggestion, which can be find in Line 242-247.

11) Lines 224-225. It is necessary to clearly distinguish between what is known about hormones in Arabidopsis and other glycophytes under salinity and available data from halophytes.

Response: Thank you for your suggestion. We have added specific plant species in Line 258-261.

12) Not all abbreviations are deciphered, for example, for Ba, CTK, Z, they are absent.

Response: Thank you for your suggestion. We have added the full abbreviations.

13) Lines 219-242. In general, the Paragraph «Secretion of Plant Hormones» is written very superficially and sparingly.

Response: We appreciate your suggestions. We have now added the related research advancement and the detailed revision in our revised manuscript can be found in Lines 253-291.

14) Lines 246-247. Not only through photosynthesis.

Response: Thank you for your comment. We have revised the sentence in Line 307-309.

15) Lines: 248-251; 255-257; 262-264; 270-272: 307-310; 319-322; 326-328. Bad English and style.

Response: Thank you for your suggestion. Linguistic mistakes have been corrected in our revised manuscript. We hope the corrections we have made can meet with your approval.

Lines: 310-313; 318-320; 326-328; 335-337; 374-378; 388-391; 396-399.

16) Lines 273-276. Not clear what was compared to what. In general. bad sentence.

Response: Thank you for your suggestion. We have modified the sentence in Line 339-342.

17) Lines 350-354. Logic seems to be absent in this sentence. It is necessary to make it clearer.

Response: Thank you for your suggestion. We have revised this sentence in Line 422-425.

18) Lines 375-377. It is not necessary that decrease in content of an intermediate in the TCA cycle indicates an inhibition of the metabolic pathway as a whole because the metabolite content depends on both its input and its output rates.

Response: Thank you for your suggestion. We have deleted that sentence.

19) Figure 2 needs improvement. The second row, marked by pink (processes) needs to make in more details. Besides, the right column (Transcriptomics...) is not a process, but a research approach, so it does not fit well to the other columns outlining processes.

Response: Thank you for your suggestion. We have modified the Figure 2.

20) Lines 400-407. Bad English. Besides, what means Regionalization? I think the authors mean Compartmentation.

Response: Thank you for your suggestion. We have modified the sentence in Line 477-487.

We have replaced the “Regionalization” to “Compartmentation”.

21) There is a mistake at the bottom of Table 2, a shift of column Function in relation to column Gene.

Response: Thank you for your suggestion. We have modified the Table.

Reviewer 2 Report

The authors propose a review manuscript entitledResearch advances on molecular mechanism of salt tolerance in Suaeda” which may be an interesting approach to understanding the various mechanisms involved in saline stress of halophytes. On the other hand, this bibliographic review can be a starting point for understand crop tolerance to salinity as   an important issue being faced by global agricultural productivity. It is well/moderate written, the illustrations including figures and tables are adequate. My general recommendation is that the manuscript deserve of some informations to be able published on international audience. I should recommend some improvements:

      1) Strengthen the link between the information in the figures and tables with the information text;

          2) In the chapter 7 "Conclusions and Perspectives" please highlight the importance of this information for the saline tolerance of traditional crop species.

1              3) The authors must checked the scientific valid names on "Plants of the World Online (Kew Science) https://powo.science.kew.org/ (e.g. S. liaotungensis synonym of S. salsa, so the valid name is S. salsa); also the authors must insert the authors in the  plant species.

              4) Line 51: "Suaeda app." must be replaced by "Suaeda spp."; Lines 57, 75, 89, etc: "S. fruticose" must be replaced by "S. fruticosa";  Line 64 "regionalize" must be replaced by  "compartmentalize"; Line 98 "translocate" mest be  repalced by "translocation"; Lines 120-122: please rephrase; Line 162: "Under high" must be replaced by ". Under high"; Lines 370 and following is the same of the lines 162 and following; Lines 246-247: please explain the meaning of the sentence "With the photosynthesis, this species has a higher protein biomass in saline soil."; Line 375: "S. corniculate" must be replaced by "S. corniculata"; Line 387: "implied" must be replaced by "suggest"; Table 1 and figure 2 must be referred to in the text before being inserted in the manuscript; The  references in Table 1 must be in numbers; Line 404: "to destruction" must be replace by "by destruction"; Line 407: "substance" must be replaced by "compounds"; "sophisticated" must be replaced by "complex"

1

Author Response

1) Strengthen the link between the information in the figures and tables with the information text;

Response: Thank you for your suggestion. We have modified the sentence and added the information about the Figure and Table in Line 505-509.

2) In the chapter 7 "Conclusions and Perspectives" please highlight the importance of this information for the saline tolerance of traditional crop species.

Response: Thank you for your suggestion. We have added the information. Which can be found in Line 519-522.

Overall, this review might enhance integrated comprehensive understanding of salt tolerance. These results may provide elite gene resources for the genetic modification of salinity-resistant crop species and improve the efficiency of saline-alkali land utilization.

3) The authors must checked the scientific valid names on "Plants of the World Online (Kew Science) https://powo.science.kew.org/ (e.g. liaotungensis synonym of S. salsa, so the valid name is S. salsa); also the authors must insert the authors in the plant species.

Response: Thank you for your suggestion. We have revised the name of “S. liaotungensis” to “S. salsa”, and added the authors of the plant species.

4) Line 51: "Suaeda app." must be replaced by "Suaeda spp.";

Response: Thank you for your suggestion. We have revised it.

Lines 57, 75, 89, etc: "S. fruticose" must be replaced by "S. fruticosa"; 

Response: Thank you for your suggestion. We have revised these words.

Line 64 "regionalize" must be replaced by "compartmentalize";

Response: Thank you for your suggestion. We have revised it.

Line 98 "translocate" mest be repalced by "translocation";

Response: Thank you for your suggestion. We have revised it.

Lines 120-122: please rephrase;

Response: Thank you for your suggestion. We have modified the sentence in Line 134-137.

Line 162: "Under high" must be replaced by ". Under high";

Response: Thank you for your suggestion. We have revised it.

Lines 370 and following is the same of the lines 162 and following;

Response: Thank you for your suggestion. We have revised it following line 162.

Lines 246-247: please explain the meaning of the sentence "With the photosynthesis, this species has a higher protein biomass in saline soil.";

Response: Thank you for your suggestion. We have revised this sentence in Line 307-309.

Line 375: "S. corniculate" must be replaced by "S. corniculata";

Response: Thank you for your suggestion. We have revised the word.

Line 387: "implied" must be replaced by "suggest";

Response: Thank you for your suggestion. We have revised it.

Table 1 and figure 2 must be referred to in the text before being inserted in the manuscript; The references in Table 1 must be in numbers;

Response: Thank you for your suggestion. We have modified the sentence in Line 505-509.

Line 404: "to destruction" must be replace by "by destruction";

 Response: Thank you for your suggestion. We have revised it.

Line 407: "substance" must be replaced by "compounds"; "sophisticated" must be replaced by "complex"

Response: Thank you for your suggestion. We have revised it.

Reviewer 3 Report

Dear authors,   

              The review article entitled "Research advances on molecular mechanism of salt tolerance in Suaeda" was submitted to Journal: Biology. The evaluation process is completed and comments, corrections, and suggestions are mentioned in the PDF file attached, 

With Best Regards,

Author Response

Reply to the Comments

Dear Mr. Shang,

Thank you very much for giving us opportunity to revise our paper (biology-1836866), and we have tried our best to revise our manuscript according to the comments and suggestions from you and referees. We hope the correction meet with approval.

We deeply appreciate your consideration of our manuscript.

Thank you for your consideration!

With kind regards,

Sincerely yours,

Wancong Yu

Responses to the referees’ comments.
1. Comments and Suggestions for Authors

This review is devoted to advances in investigations of molecular mechanisms, biochemistry, and physiology of halophytes related to Suaeda genus. In two last decades, the representatives of this genus widely used as modelling plants for study processes underlaying salinity tolerance. The appearance of such a review is certainly useful and timely. The authors were able to review the available works covering the main aspects of molecular biology, biochemistry, and physiology of the genus Suaeda and to consider the current approaches to the study of these plants. Unfortunately, the manuscript has many disadvantages, which, however, can be eliminated with a thorough revision.

The manuscript is generally superficial, lacking depth of insight, bad literature style and bad English

Below, the main and minor comments are given in one list along text.

  • Lines 65-67. Here confusion in the mechanisms of protection against the toxic and osmotic effects of salt. To avoid toxic effects cells transport ions from cytoplasm into vacuoles and by this increase osmotic pressure in vacuoles (decrease osmotic potential) what contributes to avoiding osmotic effect.

Response: Thank you for your comment. We have modified the sentence according to your suggestions. The revised sentence can be found in Line 65-68.

  • Correctly: Suaeda fruticosa, «a» must be at the end of the word

Response: Thank you for your suggestion. We have now corrected the word “Suaeda fruticose” to “Suaeda fruticosa”.

3) Lines 101-118. Legend to Figure 1. Very wordy and not very clear. In relation of ion transport, it is hardly understandable, where the authors mean the level of cell and where of whole plant. The same, about transport across plasma membrane and vacuole. In what part of the plant the content of ions rises and in what part decreases? Speaking of ion transport, it is necessary to clearly distinguish between the cellular level and the level of the whole plant.

Response: Thank you for your comments. We have revised the ions content in plant different part according to your suggestions. The revised sentence can be found in Line 113-122.

Figure legend 1 has revised as Line 101-102.

4) Figure 1. Unclear, why are CLC, VP and NHX1 related to nucleus? SOS1 and HKT1 are PM proteins, in the figure in low right part, these proteins located rather in tonoplast. The Figure and the legend to it need in clarifying and more accuracy. Speaking about Cl-/H+ antiporters of CLC family in S. altissima, it is not correctly to refer to SaCLCg and SaCLCf, because these prateins contain only one conserved glutamate. SaCLCc1 and SaCLCd are most probably Cl-/H+ antiporters.

Response: Thank you for pointing out the mistake of CLC, VP, NHX1 related to nucleus. We have revised it in Figure 1.

According to the reference 21 and 27, SsHKT1;1 and SsSOS1 are preferentially expressed in roots of S. salsa, so, they were figured in root part.

  1. Wang, W.; Liu, Y.; Duan, H. Yin.; X. Cui, Y.; Chai, W.; Song, X.; Flowers, T.; Wang, S. SsHKT1;1 is coordinated with SsSOS1 and SsNHX1 to regulate Na+homeostasis in Suaeda salsa under saline conditions. Plant Soil 2020, 449, 117–131.
  2. Liu, Q.; Liu, R.; Ma, Y.; Song, J. Physiological and molecular evidence for Na+ and cl exclusion in the roots of two Suaeda salsa Aquat. Bot. 2018, 146, 1–7.

We have revised the mistake of SaCLCg and SaCLCf in Figure 1, and SaCLCc1 [43] was added as a Cl-/H+ antiporters.

  1. Nedelyaeva, O.I.; Shuvalov, A.V.; Mayorova, O.V.; Yurchenko, A.A.; Popova, L.G.; Balnokin, Y.V.; Karpichev, I.V. Cloning and functional analysis of SaCLCc1, a gene belonging to the chloride channel family (CLC), from the halophyte Suaeda altissima (L.) Pall. Dokl. Biochem. Biophys. 2018, 481, 186–189.

5) Line 138. Betaines are not alkaloids.

Response: Thank you for pointing out this mistake. We have revised the sentence.

6) Lines 139-147. Betaines are not only osmoregulators, but they have many other functions. In paticular, it should be noted their protective function. They perform the function of osmoregulation in the cytoplasm, in vacuoles of halophytes, ions mainly increase the osmotic pressure.

Response: Thank you for your comment. We have revised the sentence according to your suggestions and deleted the section of betaines function just as Line 157-166.

7) Lines 186-187. Unclear. In response to what did activity increase in control?

Response: Thank you for your comment. We have revised the control to 0 mM NaCl treatment.

8) Lines 189 -191. Not exact wording. Due to overexpression of the A. thaliana gene or of expression of the gene from S. altissima?

Response: Thank you for your comment. We have revised the sentence in Line 218.

9) Line 201. This paragraph is about enzymes scavenging ROSs, but not about MDA.

Response: Thank you for pointing out this mistake. We have deleted the part about MDA.

10) Lines 214-215. Why inactivation? This contradicts the above sentence. Indeed, SOD activity increases in response to salt treatment and triggers cascade of the reactions leading to ROS scavenging. This paragraph needs a scheme outlining the system of ROS scavenging enzymes and nonenzymatic antioxidant substances, as well as about effects of NaCl.

Response: Thank you for your suggestion and pointing out the mistake. We have revised it and modified the sentence as your suggestion, which can be find in Line 242-247.

11) Lines 224-225. It is necessary to clearly distinguish between what is known about hormones in Arabidopsis and other glycophytes under salinity and available data from halophytes.

Response: Thank you for your suggestion. We have added specific plant species in Line 258-261.

12) Not all abbreviations are deciphered, for example, for Ba, CTK, Z, they are absent.

Response: Thank you for your suggestion. We have added the full abbreviations.

13) Lines 219-242. In general, the Paragraph «Secretion of Plant Hormones» is written very superficially and sparingly.

Response: We appreciate your suggestions. We have now added the related research advancement and the detailed revision in our revised manuscript can be found in Lines 253-291.

14) Lines 246-247. Not only through photosynthesis.

Response: Thank you for your comment. We have revised the sentence in Line 307-309.

15) Lines: 248-251; 255-257; 262-264; 270-272: 307-310; 319-322; 326-328. Bad English and style.

Response: Thank you for your suggestion. Linguistic mistakes have been corrected in our revised manuscript. We hope the corrections we have made can meet with your approval.

Lines: 310-313; 318-320; 326-328; 335-337; 374-378; 388-391; 396-399.

16) Lines 273-276. Not clear what was compared to what. In general. bad sentence.

Response: Thank you for your suggestion. We have modified the sentence in Line 339-342.

17) Lines 350-354. Logic seems to be absent in this sentence. It is necessary to make it clearer.

Response: Thank you for your suggestion. We have revised this sentence in Line 422-425.

18) Lines 375-377. It is not necessary that decrease in content of an intermediate in the TCA cycle indicates an inhibition of the metabolic pathway as a whole because the metabolite content depends on both its input and its output rates.

Response: Thank you for your suggestion. We have deleted that sentence.

19) Figure 2 needs improvement. The second row, marked by pink (processes) needs to make in more details. Besides, the right column (Transcriptomics...) is not a process, but a research approach, so it does not fit well to the other columns outlining processes.

Response: Thank you for your suggestion. We have modified the Figure 2.

20) Lines 400-407. Bad English. Besides, what means Regionalization? I think the authors mean Compartmentation.

Response: Thank you for your suggestion. We have modified the sentence in Line 477-487.

We have replaced the “Regionalization” to “Compartmentation”.

21) There is a mistake at the bottom of Table 2, a shift of column Function in relation to column Gene.

Response: Thank you for your suggestion. We have modified the Table.

  1. Comments and Suggestions for Authors

The authors propose a review manuscript entitled “Research advances on molecular mechanism of salt tolerance in Suaeda” which may be an interesting approach to understanding the various mechanisms involved in saline stress of halophytes. On the other hand, this bibliographic review can be a starting point for understand crop tolerance to salinity as   an important issue being faced by global agricultural productivity. It is well/moderate written, the illustrations including figures and tables are adequate. My general recommendation is that the manuscript deserve of some informations to be able published on international audience. I should recommend some improvements:

1) Strengthen the link between the information in the figures and tables with the information text;

Response: Thank you for your suggestion. We have modified the sentence and added the information about the Figure and Table in Line 505-509.

2) In the chapter 7 "Conclusions and Perspectives" please highlight the importance of this information for the saline tolerance of traditional crop species.

Response: Thank you for your suggestion. We have added the information. Which can be found in Line 519-522.

Overall, this review might enhance integrated comprehensive understanding of salt tolerance. These results may provide elite gene resources for the genetic modification of salinity-resistant crop species and improve the efficiency of saline-alkali land utilization.

  • The authors must checked the scientific valid names on "Plants of the World Online (Kew Science) https://powo.science.kew.org/ (e.g. liaotungensis synonym of S. salsa, so the valid name is S. salsa); also the authors must insert the authors in the plant species.

Response: Thank you for your suggestion. We have revised the name of “S. liaotungensis” to “S. salsa”, and added the authors of the plant species.

  • Line 51: "Suaeda app." must be replaced by "Suaeda spp.";

Response: Thank you for your suggestion. We have revised it.

Lines 57, 75, 89, etc: "S. fruticose" must be replaced by "S. fruticosa"; 

Response: Thank you for your suggestion. We have revised these words.

Line 64 "regionalize" must be replaced by "compartmentalize";

Response: Thank you for your suggestion. We have revised it.

Line 98 "translocate" mest be repalced by "translocation";

Response: Thank you for your suggestion. We have revised it.

Lines 120-122: please rephrase;

Response: Thank you for your suggestion. We have modified the sentence in Line 134-137.

Line 162: "Under high" must be replaced by ". Under high";

Response: Thank you for your suggestion. We have revised it.

Lines 370 and following is the same of the lines 162 and following;

Response: Thank you for your suggestion. We have revised it following line 162.

Lines 246-247: please explain the meaning of the sentence "With the photosynthesis, this species has a higher protein biomass in saline soil.";

Response: Thank you for your suggestion. We have revised this sentence in Line 307-309.

Line 375: "S. corniculate" must be replaced by "S. corniculata";

Response: Thank you for your suggestion. We have revised the word.

Line 387: "implied" must be replaced by "suggest";

Response: Thank you for your suggestion. We have revised it.

Table 1 and figure 2 must be referred to in the text before being inserted in the manuscript; The references in Table 1 must be in numbers;

Response: Thank you for your suggestion. We have modified the sentence in Line 505-509.

Line 404: "to destruction" must be replace by "by destruction";

 Response: Thank you for your suggestion. We have revised it.

Line 407: "substance" must be replaced by "compounds"; "sophisticated" must be replaced by "complex"

Response: Thank you for your suggestion. We have revised it.
